# Nitric Oxide as a Target for Phytochemicals in Anti-Neuroinflammatory Prevention Therapy

**DOI:** 10.3390/ijms22094771

**Published:** 2021-04-30

**Authors:** Lalita Subedi, Bhakta Prasad Gaire, Sun-Yeou Kim, Amna Parveen

**Affiliations:** College of Pharmacy, Gachon University, #191, Hambakmoero, Yeonsu-gu, Incheon 21936, Korea; subedilali@gmail.com (L.S.); samarpanbp@gmail.com (B.P.G.); sunnykim@gachon.ac.kr (S.-Y.K.)

**Keywords:** nitric oxide, reactive nitrogen species, neuroinflammation, neurodegeneration, phytochemicals, medicinal plants, plants derivatives

## Abstract

Nitric oxide (NO) is a neurotransmitter that mediates the activation and inhibition of inflammatory cascades. Even though physiological NO is required for defense against various pathogens, excessive NO can trigger inflammatory signaling and cell death through reactive nitrogen species-induced oxidative stress. Excessive NO production by activated microglial cells is specifically associated with neuroinflammatory and neurodegenerative conditions, such as Alzheimer’s and Parkinson’s disease, amyotrophic lateral sclerosis, ischemia, hypoxia, multiple sclerosis, and other afflictions of the central nervous system (CNS). Therefore, controlling excessive NO production is a desirable therapeutic strategy for managing various neuroinflammatory disorders. Recently, phytochemicals have attracted considerable attention because of their potential to counteract excessive NO production in CNS disorders. Moreover, phytochemicals and nutraceuticals are typically safe and effective. In this review, we discuss the mechanisms of NO production and its involvement in various neurological disorders, and we revisit a number of recently identified phytochemicals which may act as NO inhibitors. This review may help identify novel potent anti-inflammatory agents that can downregulate NO, specifically during neuroinflammation and neurodegeneration.

## 1. Introduction

Nitric oxide (NO) is a neurotransmitter with unique biological activity and is synthesized from the amino acid L-arginine [1]. Three distinct NO synthases (NOSs) facilitate synthesis of NO from L-arginine: neuronal NOS (nNOS), inducible NOS (iNOS), and endothelial NOS (eNOS). According to the localization of these NOS enzymes in smooth cells, NO synthesis occurs in macrophages, microglia, astrocytes, neurons, and endothelial cells. NO synthesized in endothelial cells and in the brain (Figure 1) exhibits different biological activities [2,3]. NO can cause endothelium-dependent vasodilation, inhibit platelet aggregation, induce immunomodulation, inflammation, and neuronal transmission in the central and peripheral nervous systems. Therefore, the major causes of neurological disorders that are usually observed following high NO production include NO-mediated immune modulation, inflammation, and neurotoxicity [3]. NO is important for cerebral blood flow and the underlying metabolisms [4], and it can also play an important role in memory and learning, mediation of nociception, modulation of neuroendocrine functions, and behavioral activity [5,6]. Various endotoxins and exotoxins such as lipopolysaccharide, tumor necrosis factor-alpha (TNF-α), interleukin-1B, estrogen, interferon gamma, hypoxic conditions, and ethanol (Table 1), can induce microglia and macrophage activation or alter homeostasis, resulting in excessive NO production which causes harm to the physiological system, the brain, or other parts of the body. In the past few decades, NO was found to play a role in various types of headaches, including primary, vascular (such as migraine), and cluster headaches [7,8], and treatment of serotonergic, antimigraine, and other neuroinflammatory disorders with NO inhibitors may provide considerable relief [9,10]. NO is involved in several pathophysiological conditions including oxidative damage, neurodegeneration, excitotoxicity, diabetic complications, platelet inhibition, altered smooth muscle relaxation, and cell death through alteration of the functions and pathways of several target proteins, as shown in Figure 2. Medicinal plants are a major reservoir for drug discovery, and the utilization of medicinal plants has been a safe and economic means of treatment of several human ailments since ancient times [11]. Before the advancement of medicine and technology, people depended on the health effects of medicinal plants for the prevention and treatment of minor and severe conditions. Additionally, at present, a large population segment must rely on medicinal plants to treat diseases, due to poverty and difficulty in procuring medicines for other reasons [12], and the WHO reported that more than 80% of the population of developing countries depend on traditional medicines [12]. Medicinal plants including functional foods are known resources for the treatment of several diseases including wounds, inflammation, diabetes, and cancer. Inflammation is one of the most frequent concomitants of disease, and increased NO production by activated microglia in neuronal cells and activated macrophages during non-neuronal inflammation is a key biomarker and a causative factor of several secondary damages. Under such conditions, the inhibition of NO production using medicinal plants or functional foods may help develop novel treatment strategies in the future [13]. In this review, we discuss useful medicinal plants and their potential to inhibit NO production by activated microglia/macrophages. These medicinal plants and phytochemicals may be thus be used to treat neuronal and non-neuronal inflammatory diseases in humans and animals (Table 2).

### 1.1. The role of NO in Various Neurological Disorders

NO is an enzymatic product of NOS and is produced by neurons and endothelial cells in the brain [14]. It is also a gasotransmitter that easily diffuses across membranes and acts as a vasodilator, neuromodulator, and inflammatory mediator, among other functions [15]. NO exhibits coordinated effects on brain functions, and substantial evidence suggests that the NO pathway is associated with neurodegenerative disorders such as Alzheimer’s disease (AD), dementia, and Lewy body dementia [14,16,17]. NO is released by endothelial cells in the vascular system during aging when cerebral blood flow decreases in the presence of vascular risk factors, resulting in microvasculopathy with impaired NO release, associated with metabolic dysfunction [14]. NO is a powerful signaling molecule that can be both protective and degenerative. The inducible gene iNOS is responsible for NO production during brain pathologies [17]. AD pathogenesis involves several key components such as cerebrovasculature, increased inflammation, and alterations in neuronal signaling [18,19]. NO is thought to be involved in neuroinflammation because it produces free radicals which affect cellular integrity due to mitochondrial damage. The mechanisms by which Aβ increases NO production remain unclear; however, oxidative stress is one of the main causes of neuronal function alterations. In AD pathology [20], NO plays a vital role in signal transduction pathways that are important for maintaining brain, vascular, immune, and muscular functions [21]. NO can exert both neuroprotective and neurotoxic effects, and studies have suggested that NO may be the cause of neurodegeneration in Parkinson’s disease (PD) [21]. NO causes excitotoxicity, inflammation, and mitochondrial dysfunction, all of which lead to neuronal death [22]. Levodopa (L-DOPA) is the first drug of choice in PD treatment, even though it does not provide long-term protection or curative effects [23]. Therefore, therapeutic intervention with NOS inhibitors may be preferable. Furthermore, NO contains a lone pair of electrons with remarkably complex functions in oxidative stress [24]. NO/iNOS has been shown to play an important role in demyelination, blood–brain barrier disruption, and oligodendrocyte and axonal damage, which occur during MS pathology [24,25]. Moreover, NO is a gas molecule that plays a key role in blood flow and inflammation during stroke pathology [26], and a link between increased NO levels in the CSF in brain injury was observed [27]. Under appropriate conditions, NO can exert protective effects, and such impacts are concentration-dependent [26]. In the case of amyotrophic lateral sclerosis, motor neuron death is a key pathology, and this process was attributed to oxidative stress due to either ROS/RNS- or NO- and glutamate-induced neurotoxicity. Recently, it has been found that NO is involved in glutamate-induced neuronal death, probably due to the reaction of NO with superoxide anions, which results in tyrosine residue nitration and inhibition of the mitochondrial system and of glutamate transporters [28]. Furthermore, increased amounts of NO were also found in cases of autism, which may suggest a correlation, and even though the etiology of autism remains unclear, oxidative stress has been reported to play an important role in its pathogenesis, which may be attributable to increase NO production [29]. Additionally, a growing body of evidence suggests that NO plays a role in synaptic plasticity, and abnormal NO signaling can be linked to a variety of neurodegenerative dysfunctions such as dementia [30]. NO is reportedly associated with neuropathic pain development [31], and significant increases in nitrate, nitrite, and NO were found in the sciatic nerve, highlighting the importance of NO as a target in neuropathic pain therapy [32]. Headache like migraines and vascular headaches are also part of this category. Although the precise cause of migraine pathophysiology remains unknown, evidence suggests a role of NO in migraines [8,33,34]. NO is involved in nociceptive processes, and it appears to be involved in pain transmission as the NO donor glyceryl trinitrate is known to cause headaches [33]. Therefore, NO inhibition may be a promising strategy for treating migraine and vascular headaches [35]. NO is crucially associated with various pathological conditions of cells; as a neuromodulator and gaseous molecule, NO is a significant effector in epilepsy. Studies strongly suggest that NO is involved in epileptic seizures [36,37], and anticonvulsants are currently used in epilepsy treatments. Similarly, a direct association of neuropathic pain and the pathogenic impact of NO was revealed; neuropathic pain is a complex interaction between peripheral axons, sensory neurons, and the central nervous system (CNS) [38]. In addition to its key role in disease pathology, NO is critical regarding complications caused by inflammatory cascades due to diabetes, AIDS, and sequelae of lupus erythematosus (SLE). Increased release of NO or NO dysfunction may be associated with diabetic neuropathy [39]. Similarly, the main reasons for the pathogenesis of diabetic retinopathy, which is a leading cause of blindness and visual disfunction, include oxidative stress, nitrosative stress, and increased NO concentrations [40]. Furthermore, the majority of AIDS patients suffer from neurological complications where NO plays a significant role in the host’s defense system [41,42]. Moreover, NO overproduction directly leads to HIV-I infection-associated dementia. The protein HIV-I Tat was observed to induce iNOS expression, which results in dementia [41]. NO peroxynitrite-mediated nitrosative stress causes reduced immune functioning and associated neurological complications in conditions such as Lyme neuroborreliosis [43]. SLE is an inflammatory disease associated with neurological symptoms, thus NO plays a crucial role in its pathogenesis. Increased levels of nitrite and nitrate have been associated with neurological symptoms [44] in the brain and after spinal cord injury (SCI). NO affects blood flow control, and iNOS and nNOS upregulation are involved in cardiovascular disease. Increased NO production has been observed in patients with SCI, which is of interest as NO affects the prevention of blood coagulation [45,46,47]. Furthermore, increased NO production after SCI results in neuronal loss due to oxidative damage [47], and excessive NO production by activated glial cells is the main cause of potentiated neuroinflammation in diverse CNS disorders [13,48]. Therefore, controlling excessive NO production may be a promising strategy for the treatment of neuroinflammatory disorders.

**Table 1 ijms-22-04771-t001:** Factors which may induce NO production in different cell types.

Stimulator	Cell Type	References
LPS	Murine microglia (BV2), primary microglia, and RAW cells	[49,50]
TNF-α	Macrophages	[51]
IL-1β	Murine N9 microglia	[52]
Estrogen	Human umbilical vein cells (HUVEC) and human neuroblastoma cells (SK-N-SH)	[53]
IFN-γ	Macrophages	[51]
Hypoxia	Rat insulinoma cell line (INS-1), rat islet cells	[54]
Ethanol	Hypothalamic neuronal cell cultures	[55]
Advanced glycation end products	BV2 cells, endothelial cells	[56]

### 1.2. Anti-Neuroinflammatory and NO-Inhibiting Activity of Lignans and Neolignans

Balanophonin: *Firmiana simplex* (Malvaceae) is commonly found in China and South Korea; in North America, it is a popular ornamental tree, and in South Korea, it is traditionally used to treat stomach disorders and diarrhea. Balanophonin, an active constituent isolated from this plant, reduces NO production and expression of PGE2, TNF-α, IL-1β, and COX2, and it downregulates the MAPK, ERK, JNK, and p38 MAPK pathways. Furthermore, it suppresses neuronal cell death by regulating PARP cleavage in N2a cells. These findings support the role of balaniphonin in inhibiting neuroinflammation [57].

Chaenomiside A: *Chaenomeles sinensis* (Rosaceae) mainly occurs in East Asian countries including China, Japan, and South Korea, and this plant’s fruit is traditionally used to treat inflammatory diseases, diarrhea, and throat ailments. The main bioactive components of this plant are chainomisides A and C which exert various pharmacological functions such as antiviral, antidiabetic, antihyperglycemic, antioxidant, and antihyperlipidemic activities. Recently, it has been shown that chainomiside A is the main bioactive constituent inhibiting NO production and exerting anti-neuroinflammatory effects [58].

Dihydrobenzo(b) furan neolignan (7R, 8S)-dehydrodiconferyl: *Clematis armandii* (Ranunculaceae) is traditionally used to treat inflammation-related diseases such as rheumatism and urinary tract infections. This plant is common in the Northern Hemisphere, and phytochemical studies have shown that it contains steroids and lignans with pharmacological properties including anti-inflammatory and anti-aging effects. Erythro-guaiac glycerol-β-conifer ether and (7R, 8S)-dehydrodiconferyl alcohol are the two bioactive constituents (dihydrobenzo(b)furan neolignans) isolated from this plant which reduce NO production and inhibit TNF-α expression in LPS-activated BV2 cells, indicating the potential use of this plant and its compounds in treating neuroinflammatory diseases [59]. (7R, 8S)-dehydrodiconferyl alcohol inhibits COX-2 and iNOS expression, suppresses NF-κB activation, and inhibits AKT, JNK, Src, and fak phosphorylation [60].

Lignanamides (melongenamide C and cannabisin F): *Solanum melongena* L (Solanaceae) is common in the subtropical and tropical regions, and its unripe fruit is commonly used as a vegetable while roots are used medicinally to cure toothache, pruritus, hemorrhoids, beriberi, and chilblains. Several pharmacological activities of this plant have been reported, including improved blood circulation and hypnotic, sedative, analgesic, and anti-inflammatory effects. Melongenamide C and cannabisin F are the two most active lignanamides of this plant and exhibit inhibitory activity against NO generation, indicating that these compounds can be used to treat neuroinflammatory diseases [61].

Lignanoside: *Selagginella tamariscina* (Selaginellaceae) is common in tropical and subtropical regions, and in folk medicine, it is used to improve blood circulation and to treat cancer, inflammatory disorders, cardiovascular diseases, and diabetes. The crude extract inhibits cellular expression of NO, iNOS, TNF-α, and NF-κB. The plant contains several active secondary metabolites, and (2R, 3S) dihydro-2-(3,5-dimethoxy-4-hydroxyphenyl)-7-methoxy-5-acetyl-benzofuran and tamariscinoside E are the two bioactive constituents that exhibit significant suppression of NO generation in LPS-stimulated RAW 264.7 cells. These findings indicate that the plant and its constituents can be a source of anti-inflammatory agents to treat inflammatory diseases, including neuroinflammation [62].

Sambucuside: *Sambucus williamsii* (Adoxaceae) is traditionally used to treat gastrointestinal disorders, bone and joint diseases, and kidney-related diseases. It shows various pharmacological and biological activities, including hypolipidemic, antiviral, antinociceptive, anticancer, antiglycemic, antifungal, and antioxidant effects. Furthermore, studies suggested that it facilitates differentiation of pluripotent stem cells into neurons. Recently, three new compounds termed sumbucuside B, C, and E were isolated from this plant, which belong to the lignan family and show promising anti-inflammatory anti-neuroinflammatory effects due to inhibition of NO production [63].

### 1.3. Anti-Neuroinflammatory and NO-Inhibiting Effects of Flavonoids

Cudraflavanone A: *Cudrania tricuspidata* Bureau (Moraceae) is traditionally used to treat various disorders as it is a rich source of bioactive chemicals with potential pharmacological effects, including anti-inflammatory, neuroprotective, and antioxidant activities. Cudraflavanone A, an active component of this plant, exerts anti-neuroinflammatory effects by preventing NO, PGE2, and iNOS production in LPS-stimulated BV2 cells, and it inactivates the NF-κB, P38, and JNK MAPK signaling pathways. In addition, this compound induces HO-1 expression through Nrf2 translocation; however, no effect on the expression of COX-2 protein was observed. These findings indicate that cudraflavanone A can be potentially used to inhibit neuroinflammation and to prevent neurodegenerative diseases [64].

Daidzein: *Glycine max* (Fabaceae) is commonly known as soybean. Several scientific studies have been conducted to examine the traditional use of this plant, including osteoporosis prevention and cholesterol-lowering, anticancer, antidiabetic, anti-inflammatory effects. After consumption of soybean, polysaccharide fatty acids are converted into docosahexaenoic acids, which are associated with various beneficial effects on the kidney and on inflammatory disorders. Daidzein, a major active isoflavone, exhibits anti-inflammatory activity in neurological disorders through the inhibition of NO production in LPS-stimulated models, and it may cure neurological disorders by inhibiting the expression of proinflammatory cytokines, chemokines, interleukin-6 mRNA, and monocyte chemoattractant protein-1 [65,66,67].

Eupatilin: *Artemisia asiatica* (Asteraceae) is traditionally used in Asia to treat inflammatory diseases, and may also be of use for curing other conditions including liver injury, pancreatic damage, and gastric ulcers by acting on the NF-κB pathways. In South Korea, it is used to treat gastric mucosal ulcers. Eupatilin, a major bioactive constituent, was suggested to exhibit various pharmacological activities such as anti-inflammatory and antioxidant effects, according to various in vitro and in vivo studies. It regulates activated microglia in transient focal cerebral ischemia and exerts beneficial effects on inflammation [68].

Genistein: Soybean is a staple food in Asian countries and a source of oil and proteins for humans and animals. It contains phytoestrogens which have been used to treat several hormone-dependent and -independent pathologies, including a variety of cancers such as breast, prostate, and colon cancer, as well as other diseases such as obesity, cardiovascular disease, Alzheimer’s disease, and osteoporosis. Genistein, a bioactive constituent, has been shown to exert anti-inflammatory effects during neurological disorders through NO inhibition, and it inhibits NF-κB, proinflammatory cytokines, and the activation of AMP kinase, thereby exerting anti-inflammatory effects [65,69].

Orobol 4′-O-β-D-apiofuranosyl-(1→6)-β-D-glucopyranoside: *Tilia amurensis* (Tiliacea) is commonly termed “bee tree” and is widely distributed in Japan, South Korea, China, and Russia. The leaves of this plant are used to treat cancer. Tea from this plant’s flowers is frequently consumed due to its sedative, diaphoretic, and antispasmodic properties. Orobol 4′-O-β-D-apiofuranosyl-(1→6)-β-D-glucopyranoside, an isoflavonoid isolated from this plant, was examined regarding its neuroprotective activity against neurodegenerative diseases, and among several secondary metabolites isolated from this plant, this compound showed the most promising inhibition effects on NO production in LPS-stimulated BV2 cells [70].

Quercetin: *Impatiens balsamina* (Balsaminaceae) is widely distributed in India, China, and South Korea. It has a long history of use for treating contact dermatitis, scalds, burns, neuralgia, and lumbago. Several pharmacological studies have supported its therapeutic potential as an antioxidant, anticancer, anti-nociceptive, anti-inflammatory, anti-anaphylactic, and anti-pyretic agent due to its high active flavonoid content. Quercetin is one of the major active secondary metabolites that has been evaluated for its neuroprotective activity. These results indicate that *I. balsamina* is a new source of an anti-neuroinflammatory natural compounds that inhibit NO production in LPS-stimulated BV2 cells. Furthermore, quercetin regulates brain nitrite levels to counteract the activity of several oxidases and elicits antidepressant and inflammatory responses [71].

Sophoraflavanone G: *Sophora alopecuroides* (Fabaceae) is traditionally used in China to treat pain, inflammation, and fever. Sophoraflavanone G is the main secondary metabolite isolated from this plant and has been shown to exhibit anti-inflammatory effects. *S. alopecuroides* has been evaluated for its neuroinflammatory activity in LPS-stimulated BV2 cell; it downregulated the expression of NO, PGE2, TNF-α, IL-1β, and IL-6. Further studies revealed the effects of this plant can be attributed to the presence of sophoraflavanone G. Additionally, sophoraflavanone G inhibits the expression of iNOS and COX-2, downregulates the JAK/STAT, P13K/AKT, and MAPK pathways, and upregulates oxygenase-1 expression. Taken together, these results suggest that sophoraflavanone G can be potentially used as an anti-neuroinflammatory agent [72].

Tangeretin: *Citrus aurantium* (Rutaceae), commonly known as bitter orange, is widely distributed throughout Asia. Medicinally, this plant is consumed as an appetite suppressant and stimulant; however, in traditional Chinese medicine, it is used to treat cardiovascular diseases, cancer, constipation, indigestion, and nausea. This plant may counteract lung cancer due to the presence of tangeretin, an active constituent. The effects of tangeretin on neuroinflammation suggest that it inhibits neurodegenerative disorders by inhibiting NO, IL-1β, COX-2, MAPK, NF-κB, IkB-α, and IKK-β in LPS-stimulated BV2 cells, thereby counteracting neurodegenerative diseases [73].

### 1.4. Anti-Neuroinflammatory and NO-Inhibiting Activity of Polyphenols

Butein: *Toxicodendron verniciflumm* (Anacardiaceae) is traditionally used as a food supplement in folk medicine to treat dyspepsia, heartburn, abdominal infection, bronchitis, arthritis, cancer, and inflammatory disorders in South Korea. Several pharmacological activities of this plant have been reported, including apoptosis, anticancer, and antioxidant activities. This plant is a rich source of secondary metabolites such as polyphenols and flavonoids which have anticancer and anti-inflammatory properties. Buetin, one of the most active constituents present in this plant, predominantly inhibits NO production in LPS-stimulated BV2 cells [74].

(4E,6E)-1-(3′,4′-dihydroxyphenyl)-7-(4″-hydroxyphenyl)-hepta-4,6-dien-3-one, and Tsaokarylone: *Dioscorea nipponica* (Dioscoreaceae) is used traditionally in South Korea to treat diabetes, neurodegenerative diseases, chronic fatigue, loss of appetite, malnutrition, and inflammatory disorders. 3,7-dihydroxy-2,4,6-trimethoxy-phenanthrene is a bioactive constituent isolated from this plant which can counteract neurodegenerative diseases. It not only induces nerve growth factor expression but also inhibits NO production in LPS-stimulated BV2 cells without cell toxicity. In addition, it predominantly suppresses elevated neurite outgrowth in N2a cells [75].

Gingerol: *Zingiber officinale* (Zingiberaceae) is a common ingredient in food and is valued for its use in treatments of various pathological conditions associated with brain aging and neurodegeneration. Ginger contains various bioactive constituents including gingerol and its dehydrating products such as shogaol. Gingerol and shogaol exert various pharmacological effects, including antioxidant, anti-inflammatory, chemopreventive, cardiotonic, and antipyretic activities. Ginger extract inhibits NO production and proinflammatory cytokines through the NF-κB pathway, and it reduces B-amyloid-induced oxidative cell death. Moreover, gingerol and shogaol produced from ginger are active, and some ingredients such as zingerone and 6-gingerol inhibit NO production, IL-6, IL-1β, TNF-α, and mRNA levels in BV2 microglial cells activated by LPS. Fresh ginger extracts also exert anti-neuroinflammatory properties due to the presence of 10-gingerol. These observations suggest the potential of ginger and its bioactive constituents regarding anti-inflammatory effects in neurodegenerative diseases by suppressing NO production [76].

Oleuropein: *Fraxinus rhynchophylla* (Oleaceae) is distributed in several areas of the world, including eastern and western France, China, India, Pakistan, Morocco, and Afghanistan. *F. rhynchophylla* is traditionally used to treat pneumonia and malaria, and oleuropein, a major bioactive constituent isolated from this plant, has been investigated to determine its anti-neuroinflammatory activity exerted through inhibition of NO and ROS production and ERK/p38/NF-κB activation. Furthermore, it inhibits Drp1-dependent mitochondrial fission induced by LPS. These findings highlight the therapeutic value of oleuropein in microglial inflammation-regulated neurodegenerative afflictions [77].

Paradol: Paradol, a biotransformed metabolite of shogaol, ameliorates neuroinflammation and CNS disorders. It markedly lowers microglial activation, TNF-α, and iNOS expression, and ultimately reduces neuroinflammation, indicating its potential for treating cerebral ischemia [78].

Salicortin: *Salix glandulosa* (Salicaceae) is commonly known as Korean King willow in South Korea. Salicortin, a bioactive salicin derivative, is the most effective constituent isolated from this plant; it exhibits inhibitory activity against NO production, and shows potency with respect to the treatment of neuroinflammation in LPS-stimulated BV2 cells [79].

Shogaol: 6-shogaol is a bioactive constituent of *Zingiber officinale* and can be potentially used to treat several CNS disorders. It inhibits NO and inducible nitric oxide synthase in LPS models, and it exhibits neuroprotective effects in animals with transient global ischemia through suppression of microglia. Furthermore, it enhances the expression of acetyltransferase and choline transport and protects neurons. These findings support the use of ginger in multitarget therapies for Alzheimer’s disease [80].

Zingerone: Zingerone, a bioactive constituent present in cooked ginger, may have anti-inflammatory effects as it significantly lowers the levels of proinflammatory cytokines and enzymes involved in cartilage degradation [81], and it attenuates LPS-induced changes in IL-1β, IL-6, TNF-α, creatinine, blood nitrogen urea, and kidney histopathology and reduces NF-κB, TLR4, MyD88, and TRIF expression. These effects indicate a suppressive role of zingerone in LPS-induced AKI by inhibiting the TLR4/NF-κB signaling pathway [82].

### 1.5. Anti-Neuroinflammatory and NO-Inhibiting Activity of Triterpenoids

Alphitolic acid: Alphitolic acid is present in leaves of *Alphitonia petriei* (*Rhamnaceae*) and also in *Ziziphus jajuba* fruit, commonly termed “jujube”, “red date”, or “Chinese date”. In China, jujube is considered one of the most valuable fruits, and it has various traditional benefits, including calming the mind and improving sleep. Jujube contains valuable secondary metabolites that exert potential pharmacological and biological effects including sedative, antioxidant, and immunological effects. The effects of aliphatic acid, an active constituent in the fruit of this plant, on NO inhibition were examined in LPS-induced and LPS- and IFN-γ-mediated microglia and macrophages’ activation and in a neuro-inflammation model, which suggested anti-neuroinflammatory properties [83,84].

Betulinic acid and coussaric acid are produced by *Diospyros kaki* Thunb., distributed in East Asian countries, and *Chaenomeles* sp. *D. kaki* leaves are typically used to produce tea. Betulinic and coussaric acid exhibit therapeutic value as anti-inflammatory agents through the inhibition of the NF-κB pathway, and they reduce NO production and prostaglandin E2, TNF-α, IL-1β, IL-6, and COX-2 concentrations. These findings indicate therapeutic value of both agents against neuroinflammation [85].

Corosolic acid and ambradiolic acid: *Betula schmidtii* (Betulaceae) is widely distributed in China, Japan, and South Korea. In traditional Korean medicine, this plant is used to treat stomach disorders. Phytochemical screening revealed that corsolic and ambradiloic acid are the two main active constituents, which exert several biological and pharmacological activities, including antioxidant, anti-inflammatory, immunomodulatory, anticancer, and hepatoprotective activities. In neuroinflammatory disorders, NO production is inhibited by these compounds without signs of cytotoxicity [86].

Faurinone: *Lindera glauca* (Lauraceae) a widely distributed shrub in China, Japan, and South Korea which is used to treat several pathological conditions such as speech disorder, abdominal pain, and paralysis. This compound may inhibit NO production without cytotoxic effects [87] as shown in Table 2.

23-hydroxybetulinic acid: *Chaenomeles speciosa* (Rosaceae) is common in East, Central, and Southwest China. Traditionally, the fruit of this plant is used to harmonize the stomach, relax tendons and muscles, and prevent several clinical conditions including migraine, neuralgia, beriberi, dysentery, cholera, and rheumatism. Due to the fruit’s traditional use, several pharmacological studies have been conducted to examine its therapeutic value. 23-hydroxybetulinic acid, a bioactive constituent isolated from this plant, exerts substantial effects against LPS-induced inflammation, suggesting its efficacy as an anti-inflammatory agent [88].

Holophyllane A: *Abies holophylla* Maxim. (Pinaceae) is used by traditional healers to treat various diseases including stomach ache, rheumatic diseases, indigestion, and pulmonary and vascular diseases. This plant is a rich source of steroids, phenols, flavonoids, lignans, and terpenoids which exert different pharmacological effects such as anti-inflammatory, antifungal, anti-bacterial, and anticancer activities. This plant is widely distributed in Russia, China, and South Korea and is considered a source of various active compounds. Recently, a novel bioactive compound, holophyllane A, was isolated, which exhibits anti-inflammatory activity through inhibition of NO production; however, further studies are required to explore mechanisms underlying its anti-inflammatory effects [89].

Ilimaquinone: *Smenospongia cerebriformis*, a marine sponge, is a prominent source of secondary constituents with potential biological and pharmacological activities with diverse structures. Several pharmacological activities have been examined to explore the potential therapeutic value of this sponge, including its anticancer and anti-depressant effects. Ilimaquinon, the main active constituent isolated from this sponge, can suppress the proliferation of multiple myeloma cells and exert anti-inflammatory effects. Recently, we determined such anti-inflammatory effects by inhibiting NO production during LPS-induced inflammation in BV2 microglia cells, suggesting the therapeutic value of this compound in neuroinflammatory diseases [90].

Maslinic acid: *Olea euroopaea* L. (Oleaceae) is the most prominent member of the genus *Olea* and the only member of its family that is consumed as food. It is most common in the Mediterranean region, and its fruits are important in the context of religion, as olives are mentioned in different holy books such as the Bible and the Quran. The therapeutic potential of the fruits is also acknowledged in traditional medicine, as they can lower uric acid, cholesterol, and blood sugar levels. Products of this plant have been used traditionally as laxatives and mouth wash and to treat diabetes, rheumatism, hemorrhoids, asthma, gastrointestinal, respiratory afflictions, and urinary tract infections. Many phenolic compounds of this plant have been investigated in the past century, and maslinic acid, an active constituent isolated from olive pomace, is associated with a reduction in the occurrence of inflammation-related diseases. Various in vitro and in vivo studies indicate that maslinic acid reduces NO production, NOS mRNA, and protein expression stimulated by LPS. It also suppresses expression of TNF-α, NF-κB, and COX-2, which suggests the therapeutic value of maslinic acid to reduce neuroinflammation [91].

Saikosaponins: *Bupleurum falcatum* L. (Umbellifers) is traditionally used to treat autoimmune diseases and chronic hepatitis. Several active secondary metabolites have been identified in this plant, including polyacetylene, flavonoids, triterpenoids, polysaccharides, and saikosaponins. Saikosaponins (saikosaponin B3, B4, and D) are considered this plant’s major active chemicals due to several pharmacological activities including immunomodulatory, antibacterial, anti-inflammatory, and antihepatoma effects. In LPS-stimulated BV2 cells of inflammation models, saikosaponins inhibit NO production and expression of inflammatory marker genes including TNF-α, IL-6, IL-1β, iNOS, and NF-κB/real. Saikosaponin B3 was found to be less cytotoxic than other saikosaponins. Taken together, these results suggest that *B. falcatum* ameliorates neuroinflammatory diseases due to the presence of saikosaponins [92].

Sesquiterpene dimer: *Artemisia argyi* H. (Asteraceae) has a long history of use in traditional Chinese medicine due to including anti-inflammatory, anticancer, antioxidant, and antidiabetic properties effects. Sesquiterpene dimer (DSF-52), a novel sesquiterpene dimer, is present in this plant and has been investigated for its anti-neuroinflammatory activity, revealing the potency of this compound. Sesquiterpene dimers inhibit NO, PGE2, TNF-α, IL-1β, GM-CSF, and MIP-1α. It also downregulates the p38 MAPK and JNK, JAK2/STAT3-dependent inflammation pathways. This suggests the potency of this compound [93].

Spathulenol: The genus *Phaeanthus* (Annonaceae) is distributed widely throughout tropical Asia. *P. vietnamensis* is an endemic species from Vietnam and has been used to treat several inflammatory diseases, including sore red eyes, diarrhea, abdominal pain, and pimples. This plant is considered important as it is used to season different food products. Spathulenol, a bioactive constituent isolated from this plant, showed considerable activity against NO production upon LPS stimulation, suggesting the potential value of this compound for treating neurodegenerative diseases [94].

### 1.6. Anti-Neuroinflammatory and NO-Inhibiting Activity of Phytoestrogens

Coumestrol: *Medicago sativa* Linn. (Fabaceae) belongs to the legumes, is sometimes referred to as “father of all foods”, and originates from Asia. It is cultivated globally for various purposes, including medicinal uses, as animal feed, and for soil improvement. This plant has a long history of being used in America, India, Turkey, Iraq, and China to treat various digestive and central nervous system disorders. Various pharmacological activities have been studied, including lowering liver cholesterol accumulation and effects on hyperglycemia and neurodegenerative menopausal symptoms. Various secondary metabolites isolated from this plant exhibit interesting nutraceutical and medicinal properties. Coumestrol is the main phytoestrogen metabolite which can inhibit neurological disorders by reducing NO production. It inhibits neurological disorders by downregulating iNOS, interferon regulatory factor-1, stat1, MCP-1, and IL-6 expression [95].

### 1.7. Anti-Neuroinflammatory and NO-Inhibiting Activity of Coumarins

Omphalocarpin: *Toddaliae asiaticae* (Rutaceae) is commonly found in China and is traditionally used to cure rheumatism, intercostal neuralgia, trauma pain, stomach disorders, cough, indigestion, and bold-circulation. This plant contains different secondary metabolites including alkaloids and coumarins with antifungal, antibacterial, anti-HIV, antiplasmodial, and anti-platelet aggregation effects. Omphalocarpin, an active constituent of this plant, inhibits the expression of proinflammatory mediators, such as NO, TNF-α, and IL-1β and downregulates COX-2 and NOS expression in LPS-stimulated BV2 cells. Omphalocarpin in *T. asiatica* is an active anti-inflammatory agent which may be used for treating neurodegenerative diseases [96].

### 1.8. Anti-Inflammatory and NO-Inhibiting Activity of other Compounds

Aucuparin and dihydromentosolic acid: *Chaenomeles speciosa* (Rosaceae) is distributed in China and South Korea, and it is traditionally used to treat common cold, hepatitis, and rheumatoid arthritis. This plant’s fruit was suggested to be a rich source of terpenoids, tannins, and flavonoids that are linked to its biological activity as an antioxidant, antimicrobial, and anti-inflammatory. Lignan and triterpenoid glycosides isolated from twigs of this plant exhibited significant anti-inflammatory activity by regulating IL-6, IL-1β, and TNF-α in LPS-induced cells. Aucuparin and dihydromentosolic acid are the two bioactive components of this plant that are responsible for suppressing NO production in LPS-induced models without cytotoxicity, suggesting that this plant and its bioactive constituents may inhibit neuroinflammation and may be a source of anti-inflammatory agents [88].

ε-Cotonefuran: *Contoneaster* sp. (Rosaceae) is a common ornamental shrub producing ε-cotonefuran which has antifungal properties. Recently, ε-cotonefuran was also confirmed to occur in *Chaenomeles sinensis* which is known to have therapeutic potential due to its anti-inflammatory effects, and the anti-inflammatory activities of this compound were examined. It exhibits substantial inhibiting effects on NO production in LPS-activated BV2 cells. These findings suggest that this compound may be a new anti-neuroinflammatory agent, which, however, requires further studies [97].

Citrusin XI: *Citrus unshiu* (Rutaceae) is traditionally used to treat fatigue, cough, bronchitis, influenza, and various cancers and to improve skin elasticity. It is primarily cultivated on Jeju Island, South Korea, and in the southern regions of Japan and China. Citrus XI, an active cyclopeptide identified in *C. unshiu* fruit, exhibits anti-neuroinflammatory effects by suppressing NO production through inhibition of iNOS and NF-κB expression in LPS-activated RAW 264.7 cells. This suggests the potential of this plant for use in treating neurodegenerative diseases [98].

Koaburaside: *Lindera neesiana* (Lauraceae) is consumed as food, and essential oil of its fruit contains large amounts of z-citral, eucalyptol, and α-pinene. Traditionally, this plant is consumed to cure tooth pain, headache, gastric disorders, and diarrhea. It is widely distributed in Myanmar, Nepal, India, Bhutan, and China. Koaburaside, a major bioactive constituent, shows anti-inflammatory activity during LPS-induced inflammation by inhibiting NO production. In addition, it inhibits inflammatory signaling proteins such as MAPK, COX-2, and INOS [99].

Sinapoyl desulfoglucoraphenin: *Raphanus sativus* (Brassicaceae), commonly known as radish, is consumed as a condiment or vegetable. This plant has a long history of being used as a Chinese medicine to stimulate digestion, bile flow, and appetite. Traditional healers use it to treat cancer, inflammation, asthma, and stomach ailments. Sinapoyl desulfoglucoraphenin is a bioactive secondary metabolite that inhibits NO production in LPS-stimulated BV2 cells. Furthermore, it can also downregulate the expression of iNOS and inhibit the tumor-inhibiting activity in HC5-15 cells. This suggests that *R. sativus* is not only useful for inhibiting neurodegenerative diseases but may be useful for treating cancer [100].

Zanthplanispine: *Zanthoxylum schinifolium* (Rutaceae) is distributed in Japan, South Korea, and China. Traditionally, it is used to improve blood circulation and to relieve pain. The fruit is used as a spice in the local cuisine of Asia. Zanthplanispine, a typical tetrahydrofuran lignin isolated from the root of this plant, inhibits NO production in LPS-stimulated RAW 264.7 cells, indicating the therapeutic value of this plant as a source of anti-neuroinflammatory agents [101].

**Table 2 ijms-22-04771-t002:** List of phytochemicals that inhibit NO production and neuroinflammation.

Scientific Name	Compound	IC50	Class	Pharmacological Target	Pharmacological Effects	References
*Abies holophylla*	Holophyllane A	12.75	Triterpenoid	NO	Anti-inflammatory Anticancer	[89]
	Holophyllane B	18.96				
*Artemisia argyi*	Sesquiterpene dimer	NA	Terpenoid	NO, PGE2, TNF-α, IL-1β, GM-CSF, MIP-1α, p38 MAPKs., JNK, JAK2/STAT3	Anti-inflammation Anticancer Antioxidant Antidiabetic	[93]
*Artemisia asiatica*	Eupatilin	NA	Flavone	NF-κB	Anti-inflammatory Neuroprotective	[68]
*Betula schmidtii*	2α-O-benzoyl-3β,19α-dihydroxy-urs-12-en-28-oic acid	4.92	Triterpenoids	NO	Anti-inflammatory	[86,102]
	2α-O-benzoyl-19α-hydroxy-3-oxo-urs-12-en-28-oic acid	9.68				
	Corosolic acid 11-Oxo-erythrodiol	12.93 16.58		NO		
	Maslinic acid	4.46		TNF-α, NF-κB, a COX-2, NO		
	Morolic acid 3-*O*-caffeate	8.62				
	Ambradiolic acid					
	Isotachioside 4-allyl-2-hydrophenyl 1-O-β-	23.9	Phenolic derivatives			
	D-apiosyl-(1 → 6)- β -D-glucopyranoside	25.3				
	Genistein 5-O-β-D-glucoside	28.8				
	Prunetinoside	34.0				
*Brassica olifera*	Sulforaphane	5.85	isothiocyanate		Anti-inflammatory Antioxidant Anti-amnesic Neuroprotective	[56,103,104]
*Bupleurum falcatum*	Saikosaponins	NA	Terpenoid	NO, TNF-α, IL-6, IL-1β, iNOS, NF-κB	Immune modulator Antibacterial Anti-inflammatory Antihepatoma	[92]
*Capsella Bursa-pastoris*	Sesquilignan glycoside	75.13	Phenolic glycosides		Anti-inflammatory	[105]
	7S, 8R, 8′R-(−)-lariciresinol-4,4′-bis-O-glucopyranoside	48.80				
	Lariciresinol4′-O-β-d-glucoside	30.70				
	(+)-Pinoresinol-β-d-glucoside	17.80				
	Salidroside	31.14				
	3-(4-β-D-glucopyranosyloxy-3,5-dimethoxy)-phenyl-2E-propanol	62.21				
	β-Hydroxy-propiovanillone 3-O-β-d-glucopyranoside	27.91				
	Coniferin	49.21				
*Chaenomeles sinensis*	3β-O-cis-feruloyl-2α,19α- dihydroxyurs-12-en-28-oic acid	37.1	Triterpenoids	NO	Anti-inflammatory	[106]
	Maslinic acid	17.8				
	2α,3α-Dihydroxyolean-12-en-28-oic acid	21.6				
	2α-Hydroxyursolic acid	47.1				
	Betulinic acid	4.5		NF-κB, NO, prostaglandin E2, TNF-α, IL-1β, IL-6, COX-2		
	Alphitolic acid	14.5				
	3-O-cis-caffeoylbetulinic	13.4				
	Ilekudinol C	25.5				
	Chaenomin	48.37				
	Berbekorin A	49.29				
	Aucuparin	50.15		NO		
	2′-Hydroxyaucuparin	38.06				
	2′-Methoxyaucuparin	28.09				
	2′,4′-Dimethoxyaucuparin	39.64				
	ε-Cotonefuran	17.78		NO		
*Chaenomeles speciosa*	Chaenomin B	283.33	Biphenyl derivatives		Anticancer Anti-inflammatory Neuroprotective	[88]
	Chaenomin A	86.02				
	2′,4′-Dimethoxyaucuparin	272.72				
	Aucuparin	20.04				
	2′-Methoxyaucuparin	22.02				
	Maslinic acid	26.96				
	Dihydrotomentosolic acid	19.41				
	Ilexgenin B,	12.72				
	Betulinic acid	2.38		NF-κB, NO, prostaglandin E2, TNF-α, IL-1β, IL-6, COX-2		
	23-Hydroxybetulinic acid and pycarenic acid	6.70				
	Pycarenic acid	67.50				
*Citrus aurantium*	Tangeretin	NA	Flavone		Anti-neuroinflammatory	[73]
*Citrus unshiu*	Citrusin XI	70	Cyclopeptide	NO, iNOS, NF-κB	Anti-inflammatory Anti-fungal Anti-bacterial Antioxidant	[98]
*Clematis armandii*	(7R, 8S)-dehydrodiconferyl alcohol	9.3	Lignan	NO, COX-2, iNOS, NF-κB, AKT, JNK, src, fak	Anti-aging Anti-inflammatory	[59,60]
*Cudrania tricuspidata Bureau*	Cudraflavanone A	22.2	Flavonoid	NO, PGE2, iNOS, NF-κB, P38, JNK MAPK	Anti-inflammatory Neuroprotective Antioxidant	[64]
*Dioscorea nipponica*	Tsaokarylone	13.36	Phenols	NO	Anti-diabetic Anti-inflammatory	[75]
	(*4E,6E*)-1-(3ʹ,4ʹ-dihydroxyphenyl)-7-(4ʹʹ-hydroxyphenyl)-hepta-4, 6-dien-3-one	14.36				
*Firmiana simplex*	4-[(1S,2R)-1,3-dihydroxy-2-[4-[(1E)-3-hydroxy-1-propenyl]-2- methoxyphenoxy] propyl]-2-methoxyphenyl β-D-glucopyranoside	59.83	Lignan glycosides	NO, PGE2, TNF-α, IL-1β, COX2, MAPKs, ERK, JNK, p38 MAPK	Anti-neuroinflammatory Anti-inflammatory Neuroprotective	[107]
	Balanophonin	10.25	Neolignan derivative			[57]
	Firmianols A	35.39	Lignan derivatives			
	Firmianols B	>500				
	(+)-Piperitol	32.65				
	(+)-Pinoresinol	25.1				
	(+)-Syringaresinol	27.53				
	Buddlenol E	19.22				
	(+)-Sesamin	26.26				
	(−)-Pinoresinol	31.1				
	(+)-7′-Methoxylariciresinol	32.99				
	(−)-5-Methoxybalanophonin	10.0				
	Buddlenol A *Threo*-(7*R*,8*R*)-	15.23				
	Guaiacylglycerol-β-coniferyl aldehyde ether	1.05				
	*Erythro*-(7*S*,8*R*)-Guaiacylglycerol-β-coniferyl aldehyde ether	0.929				
	*Threo*-Guaiacylglycerol-8-*O*-4′-sinapyl alcohol ether	9.14				
	*Erythro*-Syringylglycerol-8-*O*-4′-coniferyl alcohol ether	9.14				
	*Threo*-Guaiacylglycerol-8-*O*-4′-coniferyl alcohol ether	32.56				
	*Threo*-Guaiacylglycerol 8′-vanillin ether	47.59				
* Fraxinus rhynchophylla *	Oleuropein			NO, ROS, ERK/p38/NF-κB		[77]
*Glycine max*	Daidzein	93.15	Isoflavones	NO, AMP kinase, NF-κB	Anti-inflammatory Neuroprotective	[65]
	Daidzein	14.09				
	Genistein	137.50				
	Genistein	10.63				
	Equol	3.45				
*Impatiens balsamina*	Balsamisides A	33.33	Biflavonoid glycosides	NO	Anti-neurodegenerative	[71]
	Balsamisides B	56.86				
	Balsamisides C	39.16				
	Balsamisides D	31.02				
	Kaempferol	8.86				
	Kaempferol 3-O-β-D-glucopyranoside,	23.50				
	Kaempferol3-O-α-L-rhamnopyranosyl-(1→2)-β-D-glucopyranoside	31.73				
	Kaempferol3-O-α-L-rhamnopyranosyl-(1→6)-β-Dglucopyranoside	44.44				
	Kaempferol 3-O-β-D-allopyranoside	80.35 19.11				
	Quercetin	19.11				
	Quercetin3-O-β-D-glucospyranoide,	55.59				
	Quercetin3-O-α-L-rhamnopyranosyl-(1→6)-β-D-glucopyranoside	24.29				
	Dihydromyricetin	32.66				
*Lagerstroemia indica*	Pterospermin A	21.4	Phenolic derivative		Anti-inflammatory	[108]
	Dihydrodehydroconiferyl alcohol	14.6				
	Alatusol	35.4				
	Ficusol	36.0				
	Evofolin-B	22.0				
	Marphenol	44.9				
*Ligustrum obtusifolium*	Obtusifolisides A	33.85	Secoiridoid glycosides		LPS-activated microglia neuroinflammation	[109]
	Obtusifolisides B	5.45				
	Oleoside-11-methylester	38.67				
	Oleoside-7,11-dimethylester 10-Hydroxyoleuropein	38.89 11.17				
	Oleuropein	14.62				
	Ligstroside	61.25				
	(2″R)-2″- methoxyoleuropein	15.45				
	Neonuezhenide	14.96				
	4,4′-(1E,1′E,3E,3′E)-3,3′-(hydrazine-1,2-diylidene) bis-(prop-1-ene-1-yl-3-ylidene) bis-(2-methoxyphenol)	12.47	Azine derivative			[110]
*Lindera glauca*	Eudeglaucone	15.90	Eudesmane sesquiterpene		Anti-inflammatory	[87,111]
	(+)-Faurinone	3.67				
	3α-Hydroxycostic acid	26.48				
	Ilicic acid	14.92				
	γ-Costic acid	24.44				
	Costic acid	12.13				
*Lindera neesiana*	Koaburaside	NA	Phenolic glycoside	NO, MAPK, COX-2, INOS	Anti-inflammatory, Neuroprotection	[99]
*Medicago sativa Linn*	Coumestrol	NA	Phytoestrogen	NO, 1 (IRF-1), stat1, MCP-1, IL-6	Antidiabetic Lower liver cholesterol	[95]
*Phaeanthus vietnamensis*	(7S,8R,8′R)-9,9′-epoxy-3,5,3′,5′-tetramethoxylignan-4,4′,7-triol	65.2	Alkaloids		Anti-inflammatory	[94]
	8R,8′R-bishydrosyringenin	25.3				
	(+)-5,5’-Dimethoxylariciresinol	73.9				
	(+)-Pinoresinol	60				
	8α-Hydroxyoplop-11(12)-en-14-one	46.3				
	Spathulenol	15.7		NO		
	1αH,5βH-aromandendrane-4β,10α-diol	29.3				
	1αH,5βH-aromandendrane-4α,10α-diol10α-diol	23.0				
	1βH,5βH-aromandendrane-4α,10β-diol	22.6				
	3α,4β-dihydroxybisabola-1,10-diene	39.9				
	Nerolidol	50.8				
	(1R,2S,4S)-4-acetyl-2-[(E)-(cinnamoyloxy)]-1-methylcyclohexan-1-ol	45.7				
*Pinus koraiensis*	Koraiensides E	24.1	Diterpenoid Glycosides		Anti-inflammatory Neuroprotective	[112]
*Raphanus sativus*	Sinapoyl desulfoglucoraphenin	45.36	4-Methylthio-butanyl derivatives	NO	Anti-inflammatory Anti-proliferative	[100]
	Genistein	137.50				
	Genistein	10.63				
	Equol	3.45				
*Salix glandulosa*	Saliglandin	120.18	Salicin derivatives	NO	Anti-inflammatory	[79]
	6′-O-(Z)-p-coumaroylsalicin	31.55				
	Salicin	85.40				
	2′-*O*-acetylsalicin	123.36				
	3′-*O*-acetylsalicin	27.27				
	Fragilin	206.12				
	Trumuloidin	114.30				
	2′-*O*-(*E*)-*p*-coumaroylsalicin 2′-*O*-(*Z*)-*p*-coumaroylsalicin	29.77 25.47				
	6′-O-(*E*)-*p*-coumaroylsalicin	38.25				
	Salicortin	13.57		NO		
	2′-*O*-acetylsalicortin	14.61				
	3′-*O*-acetylsalicortin	18.27				
	6′-*O*-acetylsalicortin	22.78				
	Tremulacin	18.59				
	Cochinchiside A	23.40				
*Sambucus williamsii*	Sambucasinol B	0.9	New iridoid glycosides Lignan	NO	Anti-inflammatory Neuroprotective	[63]
	Sambucuside C	1.3				
	Sambucuside E	1.2				
	Sambucasinol A	6.82				
	Lariciresinol	72.58				
	(7αH,8′αH)-4,4′,8α,9-tetrahydroxy-3,3′-dimethoxy-7,9′-epoxylignan	>500				
	Berchemol	215.41				
	7-Hydroxylariciresinol	128.97				
	(−)-Medioresinol	45.59				
	(−)-Pinoresinol	34.25				
	7R,8S-dihydrodehydrodiconiferyl alcohol	39.97				
*Selagginella tamariscina*	(2R, 3S) dihydro-2-(3,5-dimethoxy-4-hydroxyphenyl)-7-methoxy-5-acetyl-benzofuran	32.3	Lignanoside	NO, iNOS, TNF-α, and NF-κB	Anti-inflammatory	[62]
	Tamariscinoside E	NA				
*Smenospongia cerebriformis*	Smenohaimiens A	30.13	Sesquiterpene derivatives		Anti-inflammatory	[90]
	Smenohaimiens B	28.33				
	Smenohaimiens C	>40				
	Smenohaimiens D	>40				
	Smenohaimiens E 19-Hydroxy-polyfibrospongol	24.37 24.44				
	Ilimaquinone,	10.40		NO		
	Dictyoceratin C,	>40				
	Polyfibrospongol A,	>40				
	Polyfibrospongol B	30.43				
*Solanum melongena L*	Melongenamide C	16.4	Lignanamides	NO	Anti-inflammatory	[61]
	Cannabisin F	16.2				
*Sophora alopecuroides*	Sophoraflavanone G	NA	Flavonoid	NO, PGE2, TNF-α, IL-1β, IL-6, iNOS, COX-2, JAK/STAT, P13K/AKT, MAPKs	Anti-neuroinflammatory	[72]
*Sorbus commixta*	Sorcomisides A	NA	Phenolic glycosides		Anti-inflammatory Neuroprotective Anticancer	[113]
	Sorcomisides B					
	Sorcomic acid	180.12	Fatty Acids		Anti-inflammatory Neuroprotective Anticancer	[114]
	Methyl (3S,5S)-3,5-dihydroxyhexanoate	165.03				
	(S)-(E)-4-hydroxy-2-nonenoic acid	>500				
	3(R)-hydroxyoctanoic acid	>500				
	9-Hydroxynonanoic acid	187.87				
	Methyl 9-hydroxynonanoate	196.67				
	Azelaric acid	150.22				
	Monomethyl azelate	>500				
	(9S,12S,13R)-(E)-9,12,13-trihydroxy-10-octadecaenoic acid	186.67				
	(9S,12R,13R)-(E)-9,12,13-trihydroxy-10-octadecaenoic acid	71.25				
*Tilia amurensis*	Orobol4’-O-β-D-apiofuranosyl-(1→6)-β-D-glucopyranoside	23.42	Isoflavonoid glycoside	NO	Anti-neuroinflammatory Anti-proliferative	[70]
	Pratensein-7-O-β- D-glucoside	32.23				
	Orobol 7-O-β-D-glucoside	31.85				
	Orobol 4′-O-β-glucopyranoside	>200				
	Kelampayoside A	>200				
	Osmanthuside H	25.99				
	Salidroside	35.64				
	Dihydroconiferin	158.49				
	Isotachioside	50.29				
	Tachioside	116.66				
	Koaburside	>200				
	2-Methoxyhydroquinone	58.87				
	Scopoletin	37.69				
	Scopolin	148.92				
	Fraxin	30.02				
	n-Butyl β-D-glucopyranoside	>200				
	Adenosine	>200				
*Toddaliae asiaticae*	Omphalocarpin	7.4	Phytoestrogen	NO, TNF-α, IL-1β, COX-2	Anti-inflammatory	[96]
*Toxicodendron verniciflumm*	Butein	11.68	Phenols	NO	Cytotoxic Anti-inflammatory	[74]
*Ulmus davidiana*	Fraxetin	18.72	chromane derivative		Anti-inflammatory Neuroprotective	[115]
	(+)-lyoniresinol	12.31				
	4-O-β-D-glucopyranosyl vanillic acid	21.40				
*Wasabi japonica*	Allyl isothiocyanate	20.10	Aliphatic isothiocyanate		LPS-activated microglia neuroinflammation	[116]
*Zanthoxylum schinifolium*	Zanthplanispine	36.8	Lignan	NO	Anti-inflammatory	[101]
* Zinginber officinale *	Gingerols	NA	Polyphenols	NO, NF-κB, IL-6, IL-1β, TNF-α	Anti-inflammatory Neuroprotective	[76]
	Gingerone	NA				
	Shogaol	NA		NO		
	Zingerone	NA		IL-1β, IL-6, TNF-α, NF-κB, TLR4, MyD88, TRIF		
	Paradol	NA		TNF-α, iNOS		
	Honokiol	17.0	Pyrazole analogs		Anti-inflammatory	[117]
Positive control	L-NMMA	20.53	Methyl-arginine		Anti-inflammatory	[89]

## 2. Discussion and Future Prospective

Natural products and their derivatives are rich sources of active and novel compounds that may be used against several adverse conditions including cancers, diabetes, inflammatory diseases, metabolic diseases, neurological diseases, and many others [118,119,120,121,122,123,124]. NO, a unique biological neurotransmitter, is involved in the initiation of several biological and pathological conditions, including immunomodulation, inflammation, and neuronal transmission in the central and peripheral nervous systems, which may lead to neurological disorders. Potential candidates that target the NO-interlinked signaling pathways have been screened and were identified from different natural products and their derivatives. We summarized our previous studies, and we suggest that these plants and their respective phytochemicals may be potential candidates for treating neuroinflammation when NO plays an important role as shown in (Figure 3, Figure 4 and Figure 5). Based on our evaluation, Table 3 shows that these products are more potent than standard compounds, and we elucidate several interesting facts and findings regarding these constituents and their natural resources (Table 1 and Table 2).

This review suggests that we should consider these candidates for future prospects targeting their deep signaling pathway, clinical study, safety, and formulation. For example, *Sambucus williamsii* contains sambucasinol B, the most potent candidate to use for treating neuroinflammation which has been identified and explored recently; however, no further studies have been conducted [63].

Although some potential candidates are effective anti-neuroinflammatory agents, due to their other pharmacological effects, they may not be indicated. For example, *Clematis armandii* contains Erythro-guaiac glycerol-β-conifer ether and (7R, 8S)-dehydrodiconferyl alcohol which are anti-neuroinflammatory drugs that should not be used during pregnancy as they promote urination. Therefore, caution should be exercised when considering the use of natural resources [57].

*Diospyros kaki* Thunb. has been cultivated for many years in Asian countries such as Japan, China, and South Korea, and its leaves, containing betulinic acid, have been used to produce green tea in many countries; the leaves are not toxic, which can be attributed to different bioactive compounds. Therefore, health authorities should consider this plant a part of anti-neuroinflammatory therapy [85]. Soybean is also a commonly used plant which contains a potent anti-neuroinflammatory agent that may be used to treat patients with neuroinflammatory diseases [65]. *Lindera glauca* containing faurinone is used against cancer and neuroinflammation, thus, it can be used to treat two pathological conditions at a time to reduce the economic burden on patients while promoting patient compliance [87].

Several candidate compounds and their natural sources are not only effective against neurological disorders but also against other pathological conditions. Remarkable therapeutic success has been achieved using these candidates in patients with different diseases; for example, *Sambucus williamsii*, which contains sambucasinol B, has been traditionally used to treat bone fractures and osteoporosis [63].

NO plays an important role in maintaining body functions. Excessive production of NO can lead to the initiation of various diseases and complications, including platelet inhibition, oxidative damage, cell death, diabetic complications, endothelial dysfunction, and neutrophil activation, which are interlinked with several signaling pathways and pharmacological targets such as iNOS, TNF-α, TRIF, MYD88, IL-1β, IL-6, COX-2, MAPKs, JAK/STAT, PGE2, MCR-1, ERK, and ROS. There are some natural products that have been studied only regarding their effects on NO production; however, future studies should explore the underlying mechanisms. Examples include holophyllane A, maslinic acid, aucuparin, ε-cotyfuran, spathulenol, sinapoyl desulfoglucoraphenin, saliglandin, salicortin, scambucasinol B, melongenamide C, and zanthplanispine [79,88,89,94].

Structure activity relationship (SAR) plays a vital role in predicting biological activity based on molecular structure and identifying potential bioactive constituents. SAR analysis is a powerful technology for drug discovery and provides a guideline for the synthesis of new potential compounds along with the characterization of already available molecules. Based on our review, we suggest that SAR plays an important role in biological activity. Figure 3, Figure 4 and Figure 5 show the structures of all compounds.

The inhibition effect of orobol 4ʹ-O-β- D-apiofuranosyl -(1→6)-β-D-glucopyranoside (Figure 3) on NO generation is likely due to the addition of a β-D-apiofuranosyl-(1→6)- group to the C-4 hydroxyl group through glycosylation [70]. The anti-inflammatory activity of gingerol (Figure 4) is attributed to its alkyl side chain [76]. In the case of shogaol and paradol, chain length plays an important role regarding its biological activity and pharmacokinetic properties [78,80]. The 1-hydroxy-6-oxocyclohex-2-en-1-yl) carboxylate substructure at the C-7 position of salicortin (Figure 4) may be responsible for its NO-inhibiting effect [79]. The presence of hydroxyl at the C-14 position of ilimaquinone (Figure 5) played an important role regarding its prominent anti-inflammatory effects [90]. Molecular docking studies of masilinic acid showed that due to hydrogen bonding and hydrophobic interaction, it binds to the sPLA_2-_ GV interfacial phospholipid binding site and inhibits sPLA_2-_ GV enzymatic activity in the inflammatory signaling pathway [91]. At the C-8 position, an exomethylene group was attached to the compound spathulenol, which showed strong inhibitory effects on NO production [94]. The ester group of zanthplanispine may be responsible for its strong anti-inflammatory activity [101].

Structure–activity relationship analyses revealed that a compound’s structure plays an important role regarding its biological activity. By identifying therapeutically active compounds, new derivatives can be prepared, and stronger potential agents with fewer side effects can be obtained.

In our collaborative work with other research, we aimed to identify potential NO inhibitors over the past few years. Here, we summarize many of our findings, and among them, few are high-potency agents compared to the well-known iNOS inhibitors, based on IC50 values (Table 3).

In conclusion, it is conceivable that addition of phytochemicals in the context of daily diet or consuming them as supplements can help reduce neuroinflammation or prevent its occurrence.

## Figures and Tables

**Figure 1 ijms-22-04771-f001:**
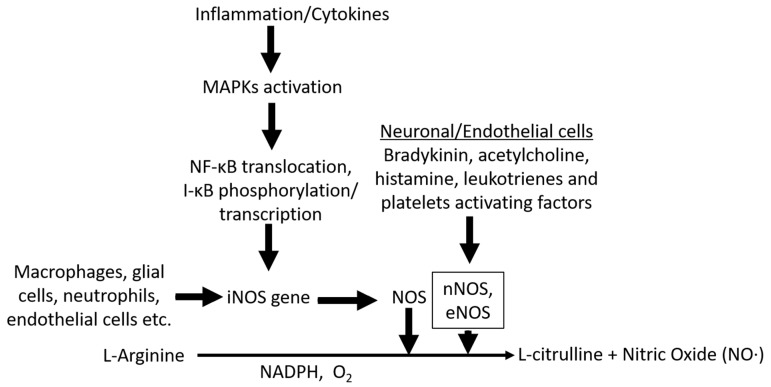
Scheme of nitric oxide (NO) synthesis MAPK activation in myeloid or glial cells can trigger NF-κB transcriptional activation, leading to expression of iNOS. iNOS is subsequently transformed to NOS, which in the presence of NADPH converts L-arginine to L-citrulline and free NO radicals. Stimulants such as bradykinin, acetylcholine, histamine, leukotrienes, and platelet-activating factors from neuronal/endothelial cells or other origins can induce expression of eNOS, and nNOS converts L-arginine to free NO radicals through the same oxidation process.

**Figure 2 ijms-22-04771-f002:**
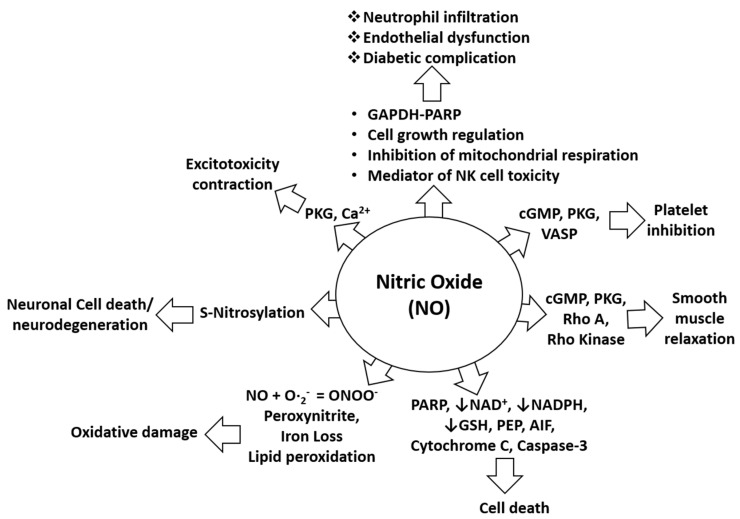
Biological role of NO in pathophysiological conditions. NO-mediated activation of cGMP, PKG, and VASP can cause platelet inhibition while NO-mediated induction of pro-apoptotic proteins such as PARP, AIF, cytochrome C, and cleaved caspase-3 can induce cell death. NO-mediated activation of cGMP, PKG, Rho A, and Rho kinase can alter smooth muscles relaxation. Inhibition of NAD, NADPH, and GSH by NO increases the probability of cell death. Lipid peroxidation caused by NO induces oxidative stress or damage. S-nitrosylation elicited by NO can cause neurotoxicity or neurodegeneration. NO-mediated induction of PKG and calcium signaling causes exitotoxicity and contraction effects. NO is also involved in neutrophil infiltration and endothelial dysfunctions through effects of mitochondrial respiration, through NK cell toxicity, and through activation of the GAPDH-PARP pathway and its functions.

**Figure 3 ijms-22-04771-f003:**
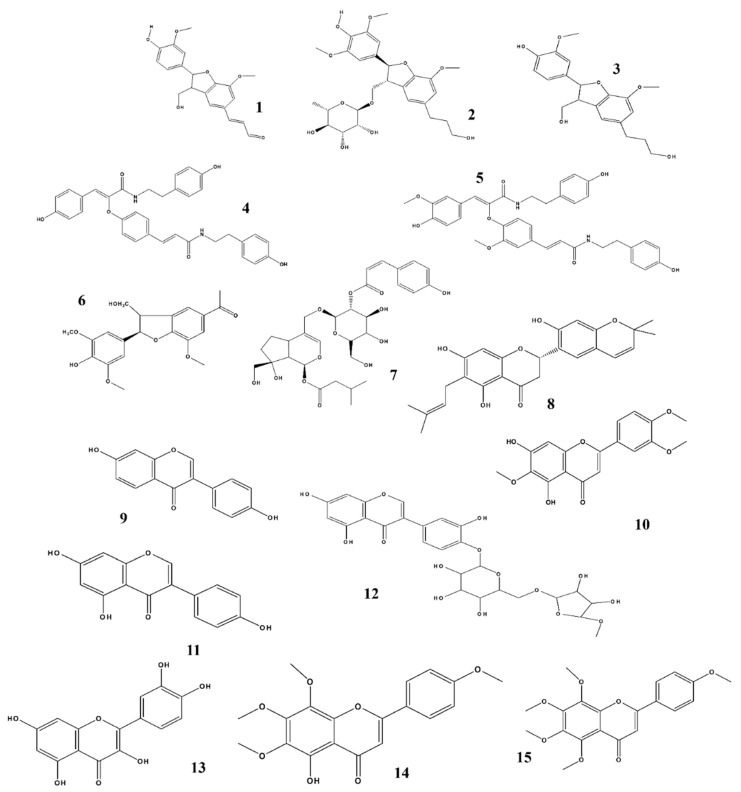
Structures of phytochemicals that can inhibit nitrite production. 1. Balanophonin_*Firmiana simplex*; 2. Chaenomiside A_*Chaenomeles Sinensis*; 3. (7R, 8S)-dehydrodiconferyl alcohol_*Clematis armandii*; 4. Melongenamide C & 5. cannabisin F_*Solanum melongena*; 6. (2R, 3S) dihydro-2-(3,5-dimethoxy-4-hydroxyphenyl)-7-methoxy-5-acetyl-benzofuran_*Selagginella tamariscina*; 7. Sambucuside_*Sambucus williamsii*; 8. Cudraflavanone A_*Cudrania tricuspidata*; 9. Daidzein _*Glycine Max*; 10. Eupatilin_*Artemisia asiatica*; 11. Genistein _*Pimpinella anisum*; 12. Orobol 4ʹ-O-β- D- apiofuranosyl-(1→6)-β-D-glucopyranoside_*Tilia amurensis*; 13. Quercetin_*Impatiens balsamina*; 14. Sophoraflavanone G_*Sophora alopecuroides*; 15. Tangeretin_*Citrus aurantium*.

**Figure 4 ijms-22-04771-f004:**
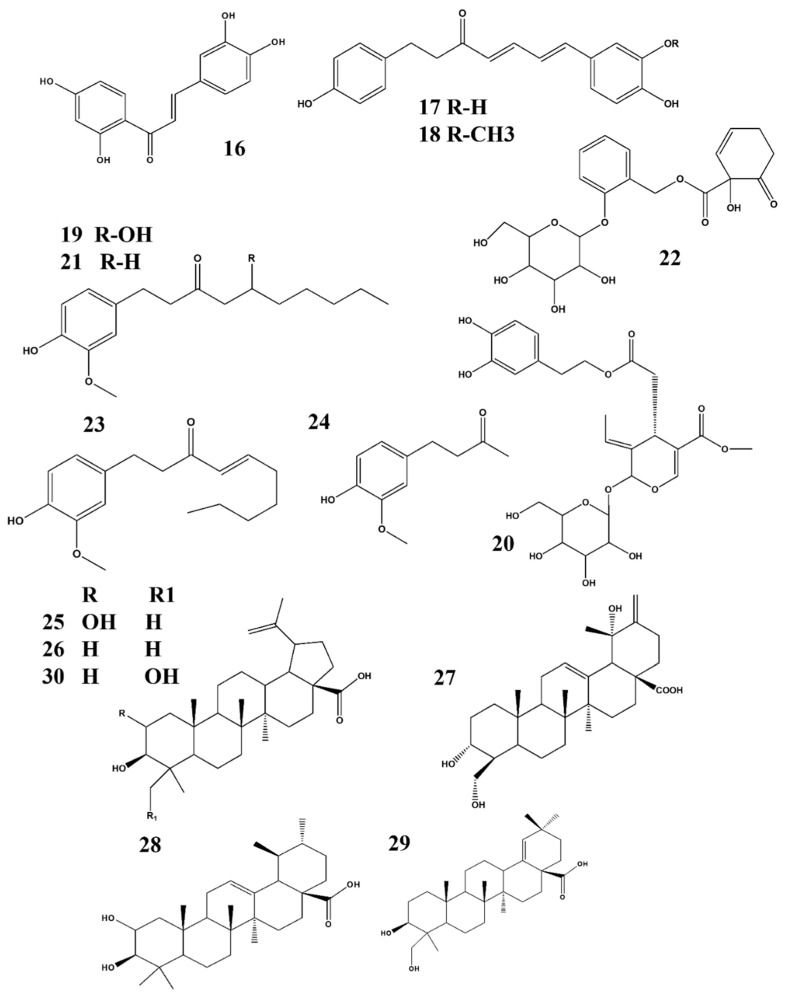
Structures of phytochemicals that can inhibit nitrite production*:* 16. Butein_*Oxicodendron vernicifluum*; 17. (4E, 6E)-1-(3′,4-dihydroxyphenyl)-7-(4″-hydroxyphenyl)-hepta-4,6-dien-3-one & 18. Tsaokarylone_*Dioscorea nipponica*; 19. Gingerol_*Zingiber officinale*; 20. Oleuropein_*Fraxinus rhynchophylla*; 21. Paradol_*Zingiber officinale*; 22. Salicortin_*Salix glandulosa*; 23. Shogaol & 24. Zingerone_ *Zingiber officinale*; 25. Alphitolic acid_*Ziziphus jajuba*; 26. Betulinic acid & 27. Coussaric acid_ *Diospyros Kaki*; 28. Corosolic acid & 29. Ambradiolic acid_ *Betula schmidtii*; 30. 23-hydroxybetulinic acid_*Chaenomeles speciose*.

**Figure 5 ijms-22-04771-f005:**
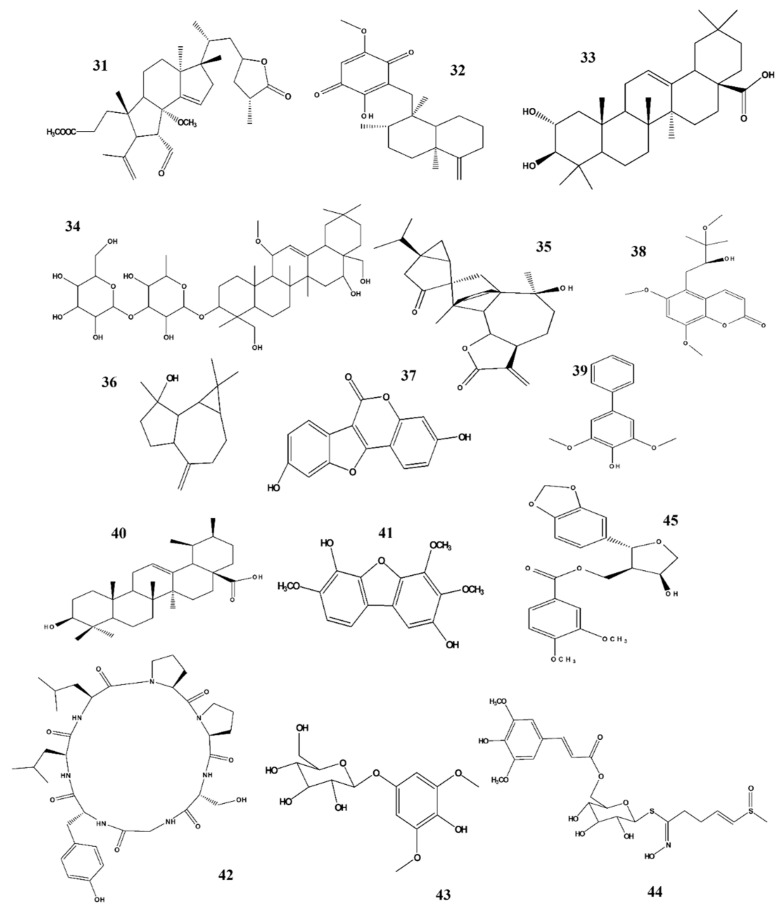
Structures of phytochemicals that can inhibit nitrite production. 31. Holophyllane A_*Abies holophylla*; 32. Ilimaquinone_*Smenospongia cerebriformis*; 33. Masilinic acid_*Olea euroopaea*; 34. Saikosaponins_*Bupleurum falcatum*; 35. Sesquiterpene dimer_*Artemisia argyi* 36. Spathulenol_*Phaeanthus veitnamensis*; 37. Coumestrol_*Medicago sativa Linn*; 38. Omphalocarpin_*Toddaliae Asiaticae*; 39. Aucuparin & 40. Dihydrometosolic acid_*Chaenomeles speciose*; 41. ε-Cotonefuran_*Chaenomeles Sinensis*; 42. Citrusin XI_*Citrus unshiu*; 43. Koaburaside_*Lindera neesiana*; 44. Sinapoyl desulfoglucoraphenin_*Raphanus sativus*; 45. Zanthplanispine_*Zanthoxylum schinifolium;* 46. (+)-Faurinone_*Lindera glauca*.

**Table 3 ijms-22-04771-t003:** Potent NO inhibitors with higher potential to inhibit NO production in LPS-activated microglia with lower IC50 than the positive control named as L-NMMA.

No.	Potential NO Inhibitor	IC50 (μM)
1.	Sambucasinol B	0.9
2.	Erythro-(7S,8R)-Guaiacylglycerol-β-coniferyl aldehyde ether	0.925
3.	Threo-(7R,8R)-Guaiacylglycerol-β-coniferyl aldehyde ether	1.05
4.	sambucuside E	1.2
5.	sambucuside C	1.3
6.	Betulinic acid	2.38
7.	Equol	3.45
8.	(+)-faurinone	3.67
9.	Maslinic acid	4.46
10.	2α-O-benzoyl-3β,19α-dihydroxy-urs-12-en-28-oic acid	4.92
11.	Obtusifolisides B	5.45
12.	23-hydroxybetulinic acid	6.7
13.	Sulforaphane	5.85
14.	Sambucasinol A	6.82
15.	1-cinnamoyltrichilinin	7.73
16.	Morolic acid 3-O-caffeate	8.62
17.	kaempferol	8.86
18.	Erythro-syringylglycerol-8-O-4′-coniferyl alcohol ether	9.14
19.	2α-O-benzoyl-19α-hydroxy-3-oxo-urs-12-en-28-oic acid	9.68
20.	Methoxy-Balanophonin	10
21.	Balanophonin	10.25
22.	Ilimaquinone	10.4
23.	Genistein	10.63
24.	Oleuropein	11.17
25.	Costic acid	12.13
26.	(+)-lyoniresinol	12.31
27.	Holophyllane A	12.74
28.	Ilexgenin B	12.72
29.	3-O-cis-caffeoylbetulinic	13.4
30.	Daidzein	14.09
31.	Alphitolic acid	14.5
32.	Dihydrodehydroconiferyl alcohol	14.6
33.	Neonuezhenide	14.69
34.	Ilicic acid	14.92
35.	Trichilinin B	15.28
36.	Eudeglaucone	15.9
37.	(2″R)-2″- methoxyoleuropein	15.45
38.	Fraxetin	18.72
39.	1-desacetylnimbolinin B	18.75
40.	L-NMMA	20.53

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
