# Peer review of "Nitric Oxide as a Target for Phytochemicals in Anti-Neuroinflammatory Prevention Therapy"

_ijms, 2021, doi:10.3390/ijms22094771_

Round 1
Reviewer 1 Report
The manuscript entitled “Nitric oxide as target for phytochemicals in anti-neuroinflammatory prevention therapy” provides a recent update on the mechanisms of NO production and its involvement in various neurological disorders, as well as tentative uses of phytochemicals as NO inhibitors. The review is well-written. However, the below revisions are recommended:
- Title: ‘as target’ should be ‘as a target’.
- Abstract Line#3: Remove ‘nitrite”.
- Section 1.4. Anti-neuroinflammatory and NO-inhibiting activity of phenols. The term ‘phenols’ should be replaced by ‘polyphenols’.
- Table 2: “Class’ is missing for some compounds. Insert the respective classes.
- Rewrite the conclusion. It does not sound meaningful.
- The manuscript requires a thorough revision of language and grammar.
- Uniformity (font and size) should be maintained in Figures 3 and 4. The authors are encouraged to follow the journal IFA.
Author Response
Thank you so much for your valuable comments. Following careful consideration of the points raised by the reviewers, we have revised manuscript.
- Title: ‘as target’ should be ‘as a target’.
Response
Title has been changed to as a target
- Abstract Line#3: Remove ‘nitrite”.
Response
Nitrile has been removed
- Section 1.4. Anti-neuroinflammatory and NO-inhibiting activity of phenols. The term ‘phenols’ should be replaced by ‘polyphenols’.
Response
The term ‘phenols’ has been replaced by ‘polyphenols’.
- Table 2: “Class’ is missing for some compounds. Insert the respective classes.
Response
Class of compounds have been added
- Rewrite the conclusion. It does not sound meaningful.
Response
Conclusion has been modified as following
In conclusion, it is suggested that addition of phytochemicals as part of our daily diet or consuming them as supplements may help to reduce neuroinflammation and act as prevention therapy. This prevention therapy foes not only improve the patient’s condition by combating neurodegenerative disorders interlinked with other complications but also reduce the cost of therapy.
- The manuscript requires a thorough revision of language and grammar.
Response
Language and grammar has been revised in revised manuscript.
- Uniformity (font and size) should be maintained in Figures 3 and 4. The authors are encouraged to follow the journal IFA.
Response
Figure 4 has been revised
Reviewer 2 Report
The review deals with phytochemicals which are able to act as inhibitors of NO, whose excessive production has been reported to be involved in several neurological disorders. The subject, which also summarizes the Authors’ findings, is very interesting and topical. The paper is well written and clearly explained. However, I would kindly suggest to check the following minor points:
-Page 1, line 5; concerning on the localization of NOS enzymes, the Authors could include smooth muscle cells
- Page 1, line 6; the sentence could be modified as follows: … and in the brain (Figure 1) exhibits different biological activities [2, 3].
- Page 1, line 7-10; please, check the construction of the sentence
-Page 2, line 6; modify as follows: altered smooth muscle relaxation
-Legend of Figure 2: a sentence related to the pathways leading to smooth muscle relaxation is missing
-Table 1; the line 6 should be deleted
-Page 10, line 16; did the Authors refer to the sesquiterpene dimer DSF-52? If so, it should be specified.
-Table 2; please, check that all IC50 values are indicated
-Table 2; the reference for Oleuropein should be added
-Page 18, line 2 from bottom; insert (figure 3)
-Page 18, line 1 from bottom; insert (figure 4)
-Page 19, line 5; insert (figure 5)
-Figure 4; the compounds number 19 and 21 are missing in the figure
-Reference nr. 125; In the last line 추계총회 및 학술대회 is reported. Do the Authors refer to the "Proceedings of the Fall International Convention of The Pharmaceutical Society of Korea"?
Author Response
Thank you so much for your valuable comments and suggestions.
Reviewer 2
The review deals with phytochemicals which are able to act as inhibitors of NO, whose excessive production has been reported to be involved in several neurological disorders. The subject, which also summarizes the Authors’ findings, is very interesting and topical. The paper is well written and clearly explained. However, I would kindly suggest to check the following minor points:
-Page 1, line 5; concerning on the localization of NOS enzymes, the Authors could include smooth muscle cells
Response
According to reviewer comments, smooth muscles cells has been added
- Page 1, line 6; the sentence could be modified as follows: … and in the brain (Figure 1) exhibits different biological activities [2, 3].
Response
Sentence has been modified according to reviewer comment.
- Page 1, line 7-10; please, check the construction of the sentence
Response
Sentence has been revised as, “NO can cause endothelium-dependent vasodilation, inhibit platelet aggregation, induce immunomodulation, inflammation, and neuronal transmission in the central and peripheral nervous systems.”
-Page 2, line 6; modify as follows: altered smooth muscle relaxation
Response
Sentence has been modified according reviewer comment
-Legend of Figure 2: a sentence related to the pathways leading to smooth muscle relaxation is missing
Response
NO-mediated activation of cGMP, PKG, Rho A, and Rho kinase can alter smooth muscles relaxation
-Table 1; the line 6 should be deleted
Response
The line 6 has been removed
-Page 10, line 16; did the Authors refer to the sesquiterpene dimer DSF-52? If so, it should be specified.
Response
It has been specified according to reviewer comment
-Table 2; please, check that all IC50 values are indicated
Response
All IC50 values are rechecked and mentioned in case missing
-Table 2; the reference for Oleuropein should be added
Response
Reference has been added
-Page 18, line 2 from bottom; insert (figure 3)
Response
Figure 3 has been added according to reviewer comment
-Page 18, line 1 from bottom; insert (figure 4)
Response
Figure 4 has been added according to reviewer comment
-Page 19, line 5; insert (figure 5)
Response
Figure 5 has been added according to reviewer comment
-Figure 4; the compounds number 19 and 21 are missing in the figure
Response
Compound number 19 and 21 have been added in figure
-Reference nr. 125; In the last line 추계총회 및 학술대회 is reported. Do the Authors refer to the "Proceedings of the Fall International Convention of The Pharmaceutical Society of Korea"?
Response
Sentence has been modified according to reviewer comment.